# Influence of Different Welding Parameters on the Morphology, Microstructure, and Mechanical Properties of 780 Duplex-Phase Steel Laser Lap Welded Joint

**DOI:** 10.3390/ma15103627

**Published:** 2022-05-19

**Authors:** Shuwan Cui, Shuwen Pang, Suojun Zhang, Yong Liao, Hongfeng Cai

**Affiliations:** 1Dongfeng Liuzhou Automobile Co., Ltd., Liuzhou 545005, China; swcui@gxust.edu.cn (S.C.); pangshuwen273437@163.com (S.P.); 888liaoyong@163.com (Y.L.); hscaihong@163.com (H.C.); 2School of Mechanical and Automotive Engineering, Guangxi University of Science and Technology, Liuzhou 545006, China; 3College of Mechanical and Vehicle Engineering, Hunan University, Changsha 410082, China

**Keywords:** laser lap welded joint, morphology, microstructure, mechanical properties, linear relationship

## Abstract

This paper attempted to establish a relationship between the morphology, microstructure and mechanical properties of a laser lap welded joint (WJ) of 780 duplex-phase (DP) steel under different welding parameters. The experimental results showed that the microstructure of the heat-affected zone (HAZ) of all the WJs were tempered martensite and equiaxed ferrite. The microstructure at the fusion zone (FZ) in all the WJs was dominated by lath martensite and ferrite, and the grain size of the FZ was larger than that in the base materials (BMs). The mechanical properties of the welded joints were tested by a universal testing machine, and the changing law of lap tensile resistance with the laser-welding parameters was analyzed. The results show that there was a linear relationship between the width of the weld and the tensile-shear forces of the weld, and the penetration of the weld had no obvious effect on the tensile-shear forces of the weld. A binary linear-regression equation was established to reveal the degree of influence of welding speed and laser power on the mechanical properties of WJs. It was found that the laser power had a greater influence on the mechanical properties of WJs than the welding speed.

## 1. Introduction

Duplex-phase (DP) steels contain both a ductile ferrite matrix and hard martensite, and have good forming, a high energy absorption rate, a low yield strength ratio and initial hardening [1,2,3,4]. It has been widely used as safety structure material for the frame and crash zones of vehicles. Welding plays a vital role in the microstructure and grain size of DP-steel-welded joints and affects the mechanical properties of WJs. Therefore, the change mechanism of the microstructure and mechanical properties of DP steel joints has attracted the attention of many scholars during the welding process.

Laser welding is a high-energy-density welding process, which has many advantages such as a high energy density, large weld penetration, small heat-affected zone and small distortion level of the workpiece after welding [5], as compared to other welding methods of DP steel, such as resistance spot welding [6,7,8,9] and gas tungsten arc welding [10]. Mei et al. [11] investigated the characteristics of CO_2_ laser overlap welding and resistance spot welding on DC56D galvanized steel used for auto bodies. The tensile-shear test results showed that the tensile-shear bearing capacity of a deep penetration is far greater than that of a resistance welding spot. Elmesalamy et al. [12] compared residual stresses in multi-pass narrow-gap laser welds (NGLW) and gas tungsten arc (GTA) welds in AISI 316L stainless steel. The residual stresses were generally 30–40% lower in magnitude for the NGLW welds in comparison to those for GTA welding. Due to the size and assembly limitations of certain structures in automobiles, laser lap welding was used in actual production. In order to prevent post-weld deformation, the laser-lap-joint process requires a larger clamping preload, which will lead to excessive residual stress. Schlather et al. [13] found that resistance spot welding requires stronger clamping than laser beam welding. Combined with the above studies, it can be found that laser welding is an effective way to improve the performance of lap joints.

Many studies have been conducted on the microstructure and mechanical performance of laser-welded joints. Bhanu et al. [14] successfully obtained a laser-welded dissimilar joint of P91 steel and Incoloy 800 HT nickel alloy. The tensile strength of the laser-welded dissimilar joint was 546 ± 6 MPa, which was lower than the base metals P91 and Incoloy 800 HT. Wang et al. [15] studied the mechanical performance and fracture behavior of 1000 DP steel laser-welded joints and found that the hardness of the FZ had the highest hardness values, which reached a maximum of more than 400 HV_0.2_. The hardness of the softening zone varied from 224 HV_0.2_ to 293 HV_0.2_, and tensile fracture occurred in the softening zone. Wang et al. [16] observed the fracture behavior of 800 DP steel laser-welded joints with different heat inputs and found that the location of the fracture gradually shifted from the FZ to the softened zone with increasing welding speeds. From the above research, it was found that there is a softening zone in the laser joint, which is prone to fracture. However, there have been few studies on laser lap welding of DP steel, and the effect of welding-process parameters on the fracture mechanism of laser lap joints is still unclear.

In this paper, 780-DP-steel laser-lap-welding experiments were performed using welding speed and laser power as independent variables. This article researched the evolution mechanism of the morphology, microstructure and mechanical properties of joints with different welding parameters. A binary linear-regression equation was established to reveal the degree of influence of welding speed and laser power on mechanical properties of WJs.

## 2. Materials and Methods

### 2.1. Materials and Welding Procedure

The laser generator of laser lap welding was controlled by a FANUC robot system with a six-axis servo motor, which is shown in Figure 1. The lap-joint-welding method was adopted. The laser was emitted by a continuous-wave laser (RFL-C6000). Pure argon was used as the protective gas with a flow rate of 15 L/min. The specific welding parameters are shown in Table 1. The 780 DP steels plates with thickness of 2 mm were selected as the BM, its mechanical properties are shown in Table 2, and its chemical composition is displayed in Table 3. 

### 2.2. Experimental Procedure

Tensile-shear tests were performed on the WJs of the 780 DP steels to investigate the mechanical properties of the laser weld. The method of the tensile-shear test followed ISO 14273. The structure and dimensions of the test specimens are shown in Figure 2. During the welding, gaskets were added to the upper and lower plates to ensure that the tensile-shear forces were coaxial in the tensile tests. For each set of parameters, two tensile-test specimens were taken for testing, and the average value of the two maximum tensile-shear forces was taken. 

After welding, the cross-section of the WJs perpendicular to the welding direction were cut. Water sandpapers 180#, 400#, 800#, 1000#, 1200#, 1500#, and 2000# were used to grind the cross-section of the WJs step by step. According to the ISO 3651-2 standard, nital solution (4 vol% HNO_3_ + 96 vol% C_2_H_5_OH) was used to etch the polished sample for 10–15 s. We rinsed the WJs with water after corrosion and then dried them with a hair dryer for observation of their microstructure. The microstructure of the WJs of 780 DP steel was observed by using an optical microscope (OM). The cross-sectional morphologies of the WJs were observed by Zeiss Axio Vert A1 metallurgical microscope, and the width of the weld and the penetration of the weld were obtained by using the software AxioVision SE64 Rel.4.9 (Carl Zeiss AG, Oberkochen, Germany).

## 3. Results and Discussion

### 3.1. The Influence of Welding Process on the Morphology of Weld

#### 3.1.1. The Influence of Welding Speeds on the Morphology of Weld

The front morphology and cross-sectional morphology of the WJs at different welding speeds are shown in Figure 3. It was observed that the FZ and HAZ gradually reduced with the increase in welding speeds. When the welding speed was 30 mm/s, the welds underwent undercut and severe continuous collapse. This is mainly because the recoil pressure and metal-liquid surface tension had not reached a good balance state in the laser welding [17]. Meanwhile, the HAZ was wide and the quality of weld forming was poor. When the welding speeds were 40 mm/s–60 mm/s, the WJs were completely penetrated and beautifully formed, and there were no pores, sagging, undercuts or other defects on the surface of the weld. It can be seen from Figure 3c,d that excessive speeds led to insufficient heat input, resulting in an unmelted area on the backside of the weld, and the FZ was narrow. 

#### 3.1.2. The Influence of Laser Power on the Morphology of Weld

The front and cross-sectional morphologies of the weld under different laser powers are shown in Figure 4. It can be observed that the HAZ gradually increased with the increase in laser power, and the upper surface of the WJs gradually changed from bumps to sags. When the laser power was 1800 W, the two plates were not welded together, and only the cross-sectional topography of the upper plate is shown in Figure 4a. The welding-process parameter of 1800 W was no longer considered on account of this phenomenon. When the laser power was 2400 W, 3000 W and 3600 W, the transition of the weld edge was continuous and smooth, no obvious pores or other welding defects were found on the surface of the welds, the cross-sections of the welds were nail-shaped, and the back of the workpiece was not completely penetrated. When the laser power was 4800 W, the FZ became smaller, concavity in weld bead profile increased, and the back side of the weld was reversely molding. The main reason for this phenomenon is that excessive energy density caused serious ablation on the front and back of the weld, which led to the serious loss of metal in the molten pool [18]. 

### 3.2. The Influence of Welding Process on the Microstructure of WJs

#### 3.2.1. The Influence of Welding Speeds on the Microstructure of WJs

The microstructure of the BM was composed of martensite and ferrite, as shown in Figure 5. The dark phase was martensite and the light phase was ferrite. The microstructure of the FZ and HAZ of the WJs were found to have significantly changed compared to the BM, as displayed in Figure 6 and Figure 7. When the welding speeds were 30 mm/s–60 mm/s, it can be seen that the HAZ consisted of tempered martensite and equiaxed ferrite that were finer than the grain of the BM. This was mainly because after the HAZ underwent welding thermal cycling, the high peak temperature did not exceed the fully austenitic temperature (Ac3), which caused carbides and alloying elements such as manganese to limit the transformation of martensite [19,20]. Meanwhile, the martensite in the HAZ was equivalent to having undergone a tempering heat treatment to refine the grains during the welding process. Figure 7 shows the OM image of the microstructure at the FZ of joints. It can be seen that the microstructure at the FZ in all the WJs was dominated by lath martensite and ferrite, and the grain size of the FZ was larger than that in the BM. Meanwhile, it is evident that the content of martensite of the WJs gradually increased with the decrease in welding speeds. This was mainly because as the welding speed decreased, the heat input increased during the cooling process, and the grains had enough time and heat to grow and form lath martensite [21].

#### 3.2.2. The Influence of Laser Power on the Microstructure of WJs

Figure 8 shows the microstructure of the HAZ of a DP 780 laser-welded joint under different laser powers. It can be found that the grain size of the HAZ was smaller than that in the BM, and the grain size of HAZ gradually increased with increasing laser power. It can be seen from Figure 9 that the most significant feature of the FZ compared to the HAZ is that the crystal grains were columnar crystals, and the degree of consistency of the crystal grain direction in the FZ became higher with the increase in laser power. This is mainly because the easiest direction of crystal growth is consistent with the direction of the maximum temperature gradient. The direction of the maximum temperature gradient gradually became consistent with the increase in laser power, which was conducive to the consistent growth of crystal grains toward the center of the molten pool to form coarse columnar crystals.

### 3.3. The Influence of Welding-Process Parameters on the Strength of Weld

The welding heat input **E** determines the melting width of the weld and penetration [22]. According to the welding-heat-input formula **E** = **P**/**V**, the welding speeds **V** and the laser power **P** are important parameters that affect the welding-heat-input value. The fracture results of the welds are shown in Figure 10, revealing that the welding seam underwent shear failure in the welding part of the two plates. The shear strength of WJs is shown in Table 4 with different welding parameters. It can be found that the tensile strength of welded joints is almost the same under different welding parameters. The relationship between the bearing capacity and the weld penetration and weld width is difficult to establish based on the tensile strength. Therefore, this paper established the relationship between the bearing capacity and the weld penetration and weld width according to the shear force. The minimum weld tensile strength of resistance welding for low-carbon steels was defined as 7400 N according to the ISO Standard ISO 14373 (resistance welding procedure for spot welding of uncoated and coated low carbon steels) [23]. Therefore, 7.4 KN was set as the standard in this study to determine good or poor welding quality.

#### 3.3.1. The Influence of Welding Speeds on the Tensile-Shear Forces of Weld

The relationship between weld width, weld penetration and tensile-shear forces under different welding-speed conditions are shown in Figure 11. It can be seen that weld penetration and weld width decreased with the increase in welding speeds, and the maximum tensile-shear forces also gradually reduced. It can be seen that the welding speed can significantly affect the tensile-shear force of the weld. When the laser power was 4200 W, the shear strength of the WJs obtained at the welding speed of 30–60 mm/s met the allowable shear strength.

#### 3.3.2. The Effect of Laser Power on the Tensile-Shear Forces of Weld

The relationship between weld width, weld penetration and tensile-shear forces under different laser-power conditions was shown in Figure 12. It can be seen that the width of the weld increased with the laser power increasing, and the tensile-shear forces also increases linearly. When the laser power was 4800 W, the penetration of weld got smaller. But the tensile-shear force didn’t decrease but increased with the decrease in the penetration of weld. It came to conclusions that the size of the tensile-shear forces mainly depends on the width of the weld, and the penetration of weld has no obvious effect on the tensile-shear forces of the weld. This relationship fully explains that the welded joint fails at the overlap of the welded plate, which is consistent with the result analysis of the shear fracture of the welded plate in the above tensile test.

### 3.4. Tensile-Shear Estimation Model Using Regression Analysis

Laser welding is a process of multi-factor interaction. Various process parameters are both interrelated and mutually restricted, and a large number of process tests, analyses and experience are needed to select the best welding-process parameters [24]. In order to obtain good-quality welds, the selection of welding-process parameters is very important. Therefore, a binary linear-regression equation was established to explore the degree of influence of each welding-process parameter on the quality of the weld. The binary linear-regression equation was assumed as:$$\hat{y}=a_{0}+a_{1}x_{1}+a_{2}x_{2}$$
where $\hat{y}$ is tensile-shear force, $x_{1}$ is the welding speed, x_2_ is the laser power, a, $a_{1}$ and $a_{2}$ are constant.

Then, the equation was established by the least-square method as
$$q=\overset{n}{\underset{j=1}{\sum }}{\left(y_{j}-{\hat{y}}_{j}\right)}^{2}$$
$$\;\bar{y}=\frac{1}{n}\overset{n}{\underset{j=1}{\sum }}y_{j},\;\bar{x_{k}}=\frac{1}{n}\overset{n}{\underset{j=1}{\sum }}x_{kj}\left(k=0,1,2\right)$$
$$\frac{\partial q}{\partial a_{k}}=0\;\left(k=0,1,2\right)$$

The resulting linear equation:$$\hat{y}=-0.183x_{1}+0.003x_{2}+9.709$$

Meanwhile, in order to verify the feasibility of the curve, the significance test (F test) was applied and the calculation formula (1) is: (1)$$F=\;\;\frac{\sum^{\;}{\left(\hat{y}-\bar{y}\right)}^{2}/k}{\sum^{\;}{\left(y-\hat{y}\right)}^{2}/\left(n-k-1\right)}=35.48>F_{0.01}\left(2,\;5\right)=13.27$$
where $\bar{y}$ is the average value of tensile-shear forces, k is the number of independent variables, n is the number of trials.

The value of F was larger than $F_{0.01}\left(2,\;5\right)$, indicating that that the curve with this equation was highly significant and reliable. It can be seen from the obtained linear equation that the shear force gradually increases with the increase in laser power and the tensile-shear forces gradually decrease with decrease in the welding speed, which is consistent with the experimental results. This result proves the accuracy of our experiment. To evaluate the effect of each welding parameter on the tensile-shear force, the regression coefficients need to be converted into standard regression coefficients. This is mainly because laser power and welding speed have different units, so we cannot simply use the regression coefficient to judge the priority of the factors. In order to eliminate the influence of different units, Equation (2) was applied.
(2)$$b_{i}^{*}=b_{i}\sqrt{\frac{\sum_{i=1}^{2}{\left(x_{i}-\bar{x}\right)}^{2}}{\sum_{i=1}^{2}{\left(y_{i}-\bar{y}\right)}^{2}}}$$
where $b_{i}$ is regression coefficient of each welding parameter, and $b_{i}^{*}$ is the standard regression coefficient of each welding parameter.

The values of the standard regression coefficients of each weld parameter are listed in Table 5. It can be concluded that the laser power had a greater influence on the weld quality than the welding speed.

## 4. Conclusions

(1) When the welding power was 4200 W, when the welding speed was 30 mm/s or 40 mm/s, or when the welding power was 1800 W or 4800 W, the morphology of WJs did not meet the welding requirements. The forming morphology of WJS under other welding parameters were good.

(2) The grain size of the HAZ of all the WJs was smaller than that in the BM. The grain size of the FZ was larger than that in the BM, and the microstructure at the FZ in all the WJs was dominated by lath martensite and ferrite.

(3) The size of the tensile-shear forces mainly depended on the width of the weld, and the penetration of weld had no obvious effect on the tensile-shear forces of the weld.

(4) There was a linear relationship between welding speed, laser power and tensile-shear forces, and we found that the laser power had a greater influence on the mechanical properties of the WJs than the welding speed.

## Figures and Tables

**Figure 1 materials-15-03627-f001:**
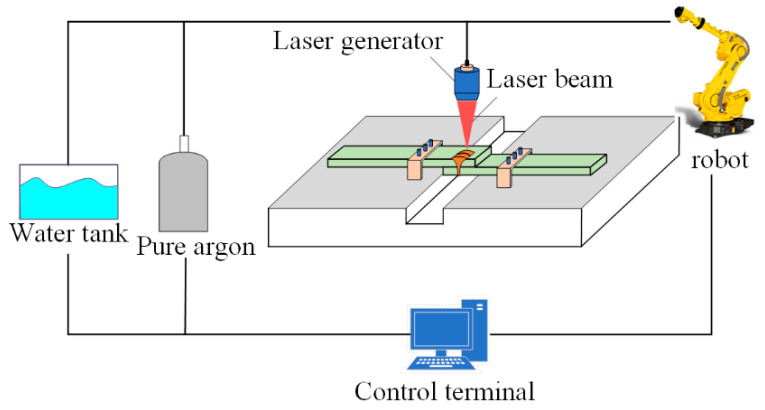
The schematic diagram of laser welding.

**Figure 2 materials-15-03627-f002:**
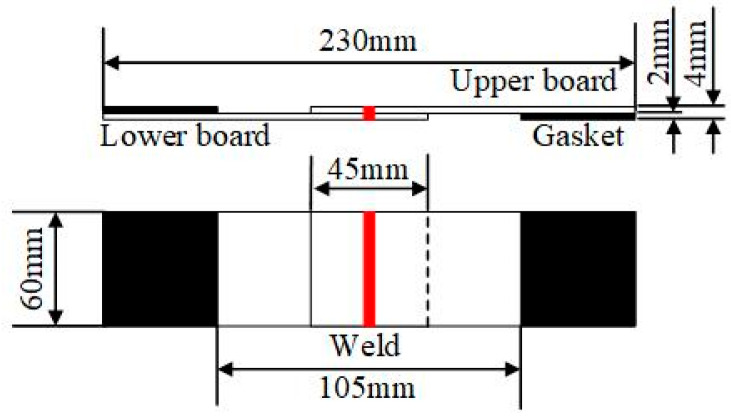
The schematic diagram of lap structure.

**Figure 3 materials-15-03627-f003:**
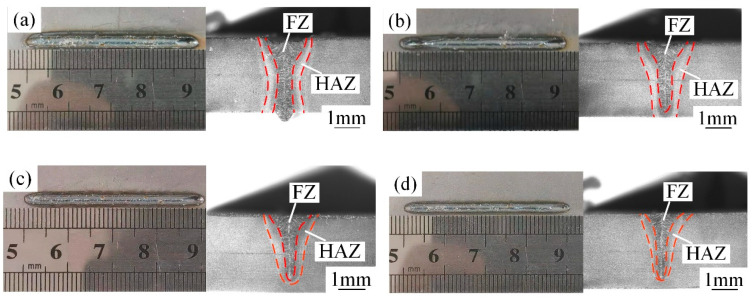
The front morphology and cross-sectional morphology of the weld under different welding speeds (mm/s): (**a**) 30, (**b**) 40, (**c**) 50 and (**d**) 60.

**Figure 4 materials-15-03627-f004:**
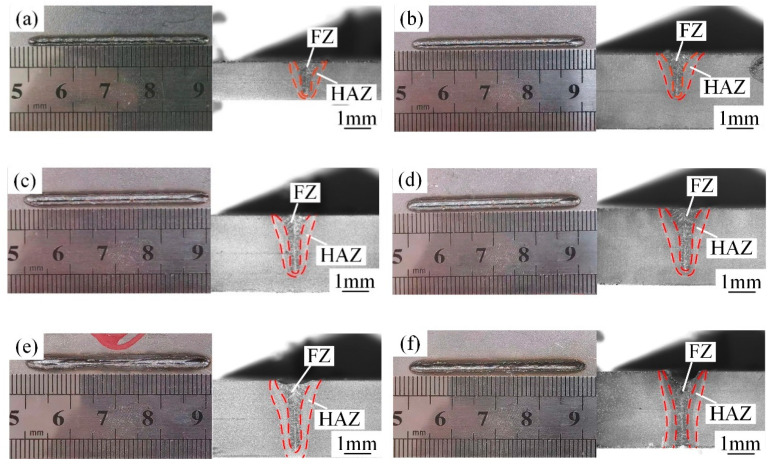
The front morphology and cross-section morphology of the weld under different laser powers (W): (**a**) 1800, (**b**) 2400, (**c**) 3000, (**d**) 3600, (**e**) 4200 and (**f**) 4800.

**Figure 5 materials-15-03627-f005:**
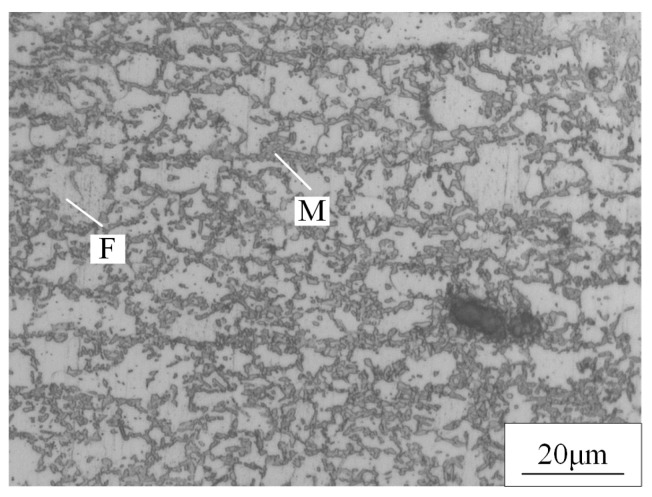
The microstructures of DP 780 BM containing martensite (M) and ferrite (F).

**Figure 6 materials-15-03627-f006:**
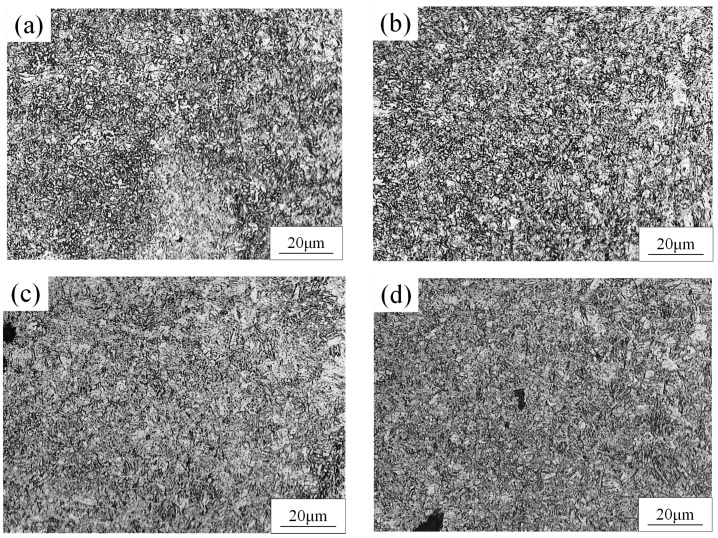
The microstructure of HAZ of DP 780 laser-welded joint under different welding speeds (mm/s): (**a**) 30, (**b**) 40, (**c**) 50, and (**d**) 60.

**Figure 7 materials-15-03627-f007:**
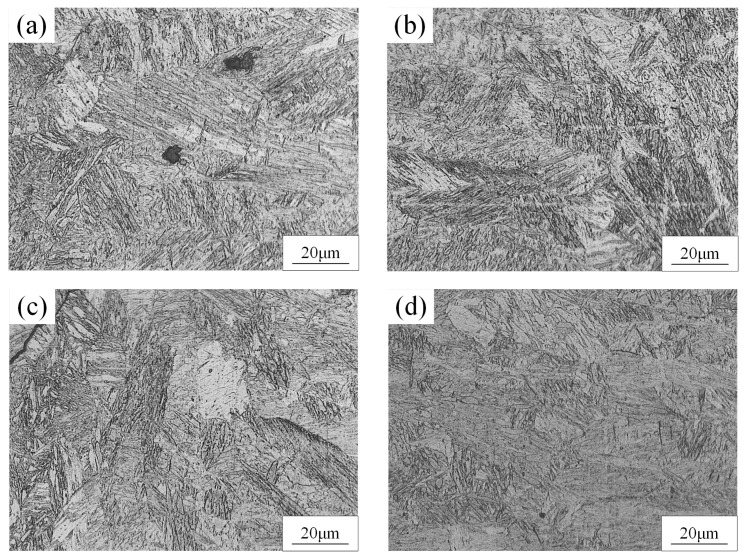
The microstructure of FZ of DP 780 laser-welded joint under different welding speeds (mm/s): (**a**) 30, (**b**) 40, (**c**) 50, and (**d**) 60.

**Figure 8 materials-15-03627-f008:**
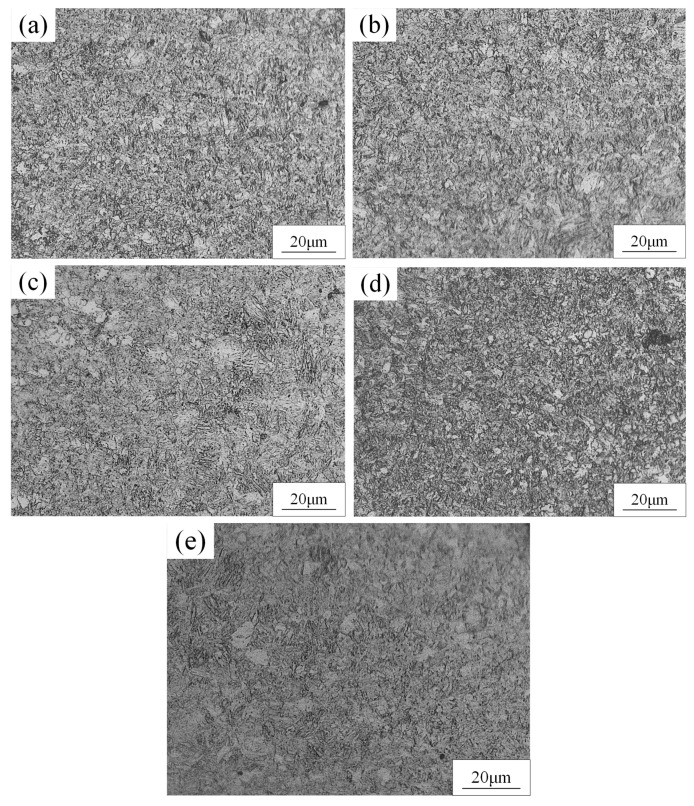
The microstructure of HAZ of DP 780 laser-welded joint under different laser powers (W): (**a**) 2400, (**b**) 3000, (**c**) 3600, (**d**) 4200, and (**e**) 4800.

**Figure 9 materials-15-03627-f009:**
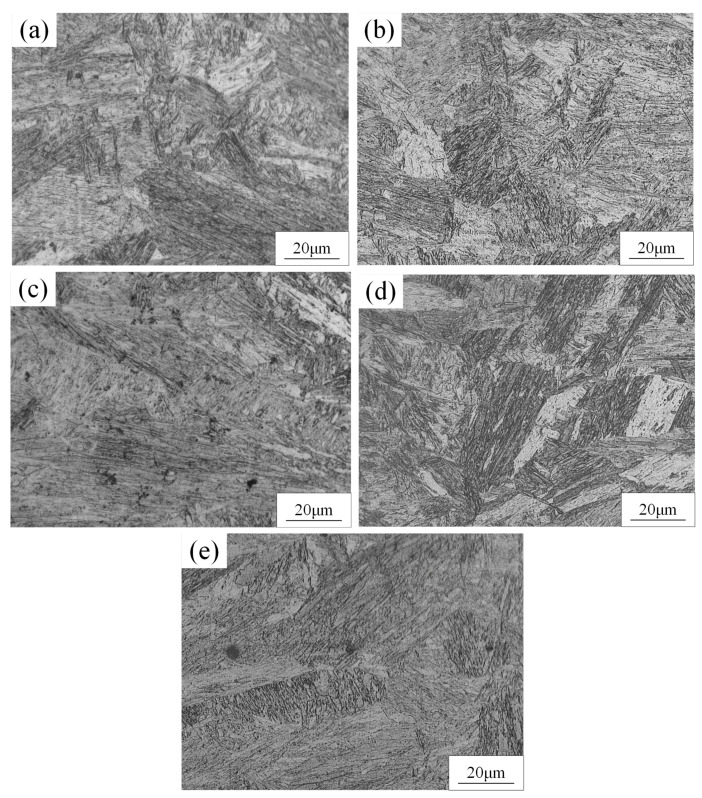
The microstructure of FZ of DP 780 laser-welded joint under different laser powers (W): (**a**) 2400, (**b**) 3000, (**c**) 3600, (**d**) 4200, and (**e**) 4800.

**Figure 10 materials-15-03627-f010:**
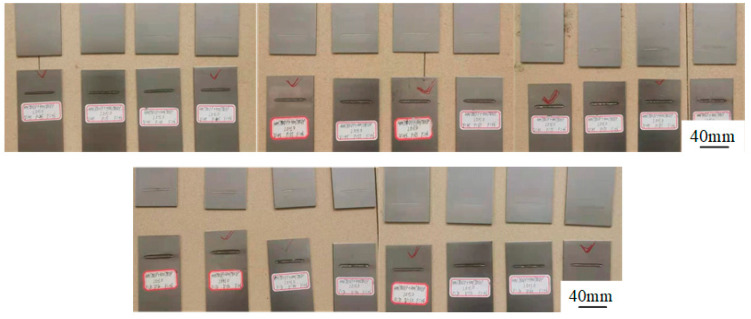
The weld-fracture results under various welding-process parameters.

**Figure 11 materials-15-03627-f011:**
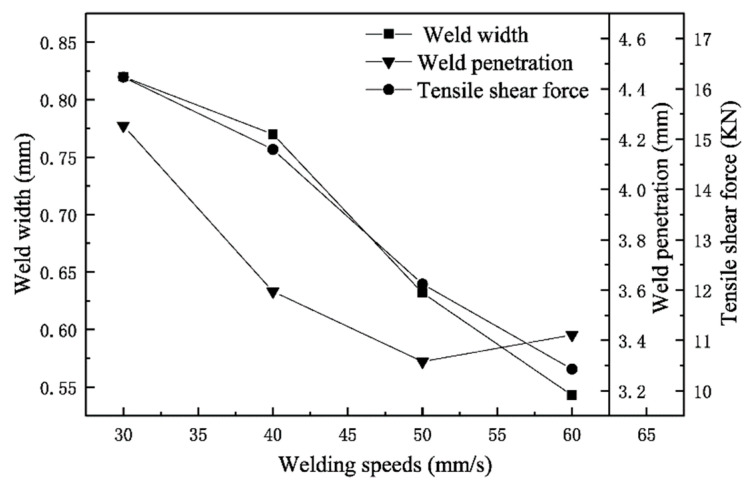
The schematic diagram of weld penetration, weld width and tensile-shear force changes at different welding speeds.

**Figure 12 materials-15-03627-f012:**
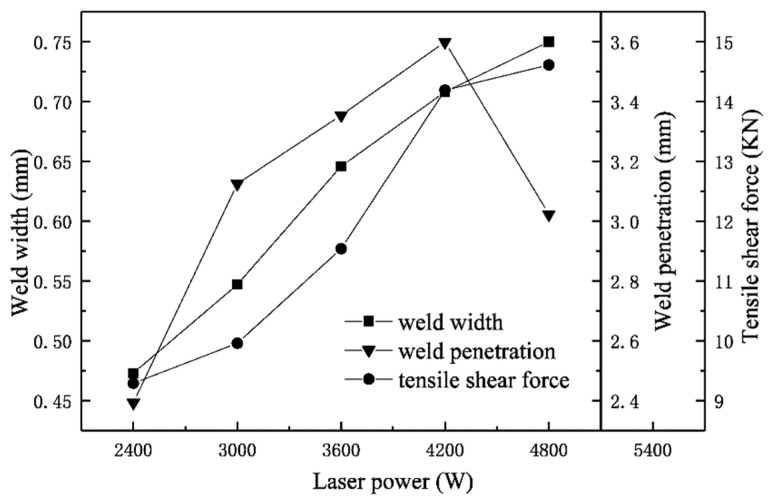
The schematic diagram of weld penetration, weld width and tensile-shear force changes at different laser powers.

**Table 1 materials-15-03627-t001:** Welding parameters.

Test	Defocusing Amount (mm)	Laser Power (W)	Welding Speed (mm/s)
1	6	4200	30
2	6	4200	40
3	6	4200	50
4	6	4200	60
5	6	1800	40
6	6	2400	40
7	6	3000	40
8	6	3600	40
9	6	4800	40

**Table 2 materials-15-03627-t002:** Mechanical properties of 780 DP steel.

Yield Strength R_eL_/MPa	Tensile Strength R_m_/MPa	Elongation after Breaking A(%)
500~650	≥780	≥10

**Table 3 materials-15-03627-t003:** Chemical composition of 780 DP steel.

Element	C	Si	Mn	P	S	Fe
**Wt.%**	≤0.2	≤0.8	≤2.5	≤0.035	≤0.2	Bal.

**Table 4 materials-15-03627-t004:** The shear strength of WJs with different welding parameters.

Test	Laser Power (W)	Welding Speed (mm/s)	Shear Strength (MPa)
1	4200	30	-
2	4200	40	468.641
3	4200	50	467.737
4	4200	60	468.49
5	1800	40	479.039
6	2400	40	444.108
7	3000	40	435.513
8	3600	40	488.666
9	4800	40	474.959

**Table 5 materials-15-03627-t005:** Standard coefficient for multiple linear-regression model.

Fator	Regression Coefficients (*b_i_*)	$\mathrm{Standard}\;\mathrm{Regression}\;\mathrm{Coefficients}\;(b_{i}^{*})$
Laser Power	0.003	0.834
Welding Speed	−0.183	−0.634

## Data Availability

Not applicable.

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
