# Peer review of "Influence of Different Welding Parameters on the Morphology, Microstructure, and Mechanical Properties of 780 Duplex-Phase Steel Laser Lap Welded Joint"

_materials, 2022, doi:10.3390/ma15103627_

Round 1
Reviewer 1 Report
Dear authors in order to meet the requirements of the journal please improve the comments listed:
In the subject line of the article, I would recommend changing
from "Influence of Different Welding Processes" to ""Influence of Different Welding Parameters"
The abstract is correct
The keywords are chosen correctly
Introduction: Authors referencing articles do not give any data to base their conclusions on.
Materials and Methods is acceptable.
Too general description of the photo 4 lack of highlighting the important elements on the photos.
Please indicate on the diagram the place where photos 5,6,7,8 and 9 were taken.
Photos 5,6,7,8 and 9 have no scale (only value is written).
Please add an additional graph with the static tension curve of the 3 types of samples used in Figures 11 and 12. The graph should describe the course of stress as a function of strain. Please mark the yield point on it. In construction materials, the yield point is the determinant of the chosen stress level.
In point 3.4 wrong formulae are written:
∑?=??+?1∑?1+?2∑?2
∑?1?=?∑?1+?1∑?12+?2∑?1?2
∑?2?=?∑?2+?1∑?1?2+?2∑?22
In the introduction, the authors write that mechanical properties will be studied.
The article does not include any results from static strength tests.
Author Response
Dear reviewer,
Thank you for your comments concerning our manuscript. These comments are all valuable and very helpful for revising and improving our paper, as well as the important guiding significance to our researches. We have studied these comments carefully and have made correction, which we hope to meet with your approval. The detailed response to reviewer is in the attachment.
Kind regards,

Reviewer 2 Report
- Please reconsider the article title; only laser process was applied with different parameters so title „different welding process“ is not convinient. Proposal is „different welding parameters“.
- Correspondence author e-mail is missing.
- Please define „ properties of WJs“.
- Which type of laser was used: fiber? Please define in order to enable repeatibility of the experiment.
- Please define „defocusing amount“, is this the focus size?
- Please check the position of the pictures (centered)?
- ISO 14373 or ISO 147273 was applied for shear strength?
- In conclusion, please define the relationship between weld width and shear strength expressed in N/mm2 to quantify influence of the microstructure!
Author Response

(The authors gave the same response as above.)

Reviewer 3 Report
Review report on the topic ‘Influence of Different Welding Processes on the Morphology, Microstructure, and Mechanical Properties of 780 Duplex Phase Steel Laser Lap Welded Joint’. Comments are listed below:
- Strengthen the abstract section. Add the key conclusion of the works in the last two lines of the abstract section. Remove the unnecessary information.
- Discuss the novelty of the work in respect of the application why laser joining was preferred to make the lap joint.
- There are numerous spelling and grammatical errors. Please revise the manuscript thoroughly. Sentences are also not complete and references are also cited in a rough manner.
- Try to make a bridge between current and previously published work and specify the gap area and objective of the work. Also, discuss the major process used for such type of joining and the problem associated with it. Also refer to some recently published work on laser joining: https://doi.org/10.3390/ma14195876;
- Provide the image of the experimental setup with good quality. Also, add the image of the welded pipe produced.
- Provide the scale for each macrograph.
- With an increase in laser power, concavity in weld bead profile was observed to be increased; why?
- Instead of optical, add good quality SE image.
- 6 –Fig. 9, quality is very poor and difficult to observe any phase from that optical image. Either improve the quality or replace it with an SE image.
- Increase the size and quality of Fig. 10.
- The results are ok but the discussion section is very poor. It looks like a technical report instead of a technical article. Improve the discussion section and add more references in support of the results: https://doi.org/10.1016/j.ijpvp.2022.104629;
- The work is good, but the technical discussion and introduction section needs improvement. Paper can be accepted after following minor corrections.
Author Response

(The authors gave the same response as above.)

Round 2
Reviewer 1 Report
Dear authors, I accept the corrections made
Reviewer 2 Report
Thank you for revisions made, hope this will help better understanding and quality of the paper.